# Weaknesses in primary health care favor the growth of acquired syphilis

**Marquiony Marques dos Santos**[1,2,3]*, **Tatyana Maria Silva de Souza Rosendo**[1,4], **Ana Karla Bezerra Lopes**[1,2,3], **Angelo Giuseppe Roncalli**[1,4], **Kenio Costa de Lima**[1,2,4]

**1** Universidade Federal do Rio Grande do Norte (UFRN), Natal, RN, Brazil, **2** Postgraduate Program in Health Sciences, Universidade Federal do Rio Grande do Norte (UFRN), Natal, RN, Brazil, **3** Doctoral Student in the Postgraduate Program in Health Sciences, Universidade Federal do Rio Grande do Norte (UFRN), Natal, RN, Brazil, **4** Postgraduate Program in Public Health, Universidade Federal do Rio Grande do Norte (UFRN), Natal, RN, Brazil

* marquiony@gmail.com

**Data Availability Statement:** All files are available from the database https://doi.org/10.6084/m9.figshare.12844727.

**Funding:** This research was funded by the Ministry of Health, and the Laboratório de Inovação

## Abstract

Acquired syphilis is a sexually transmitted infection that affects the general population and has been growing in recent years in many countries. A study was developed aiming to analyze the trends of acquired syphilis associated with sociodemographic aspects and primary health care in Brazil, in the period from 2011 to 2019. This study used secondary data from the national notification systems of the 5570 Brazilian cities and a database of 37,350 primary health care teams, as well as socioeconomic and municipal demographic indicators. The trends of acquired syphilis at the municipal level were calculated from the log-linear regression, crossing them with variables of primary health care and sociodemographic indicators. Finally, a multiple model was built from logistic regression. 724,310 cases of acquired syphilis have been reported. In primary care units, 47.8% had partial coverage and 74.1% had health teams with poor or regular scores. 52.6% had rapid test for syphilis partially available. Male and female condoms are available in 85.9% and 62.9% respectively and 54.4% had penicillin available in the health facility. The increase in trends of acquired syphilis was associated with better availability of the rapid test; lower availability of male condoms; lower availability of female condoms; lower availability of benzathine penicillin; partial coverage of the teams in primary health care; limited application of penicillin in primary health care; higher proportion of teams classified as Poor/Regular in primary health care; higher proportion of women aged 10 to 17 years who had children; higher HDI; higher proportion of people aged 15 to 24 years who do not study, do not work and are vulnerable; and population size with more than 100,000 inhabitants. The following variables remained in the multiple model: not all primary health care teams apply penicillin; higher proportion of primary health care teams with poor/regular scores; population size >100000 inhabitants; partially available female condom. Thus, the weakness of primary health care linked to population size may have favored the growth of the acquired syphilis epidemic in Brazilian cities.

Tecnológica em Saúde (LAIS/HUOL/UFRN)
process number 732017, Project: "Syphilis No."
The funders had no role in study design, data
collection and analysis, decision to publish, or
preparation of the manuscript.

**Competing interests:** The authors have declared
that no competing interests exist.

## Author summary

Acquired syphilis is a sexually transmitted infection that continues to impact health services around the world. For decades, studies and public health policies in the fight against syphilis have focused on syphilis in pregnant women and congenital, mainly because of their importance in the health of women and children. However, the behavior of the epidemic in people aged over 13 years shows that the epidemic is comprehensive and challenging. The exponential increase in the syphilis epidemic in the general population is one of the most neglected in many parts of the world. Primary health care plays a fundamental role in the control of syphilis, but few studies have sought to analyze the impact of primary health care on the growth of acquired syphilis. Our work used two large databases of the national notification systems of the 5570 Brazilian cities and a database of 37,350 primary health care teams, as well as municipal sociodemographic indicators. Our results demonstrated several indicators that influence the growth of acquired syphilis, especially the most neglected, which can guide more effective strategies to fight the syphilis epidemic in several countries.

## Introduction

Syphilis is a sexually transmitted infection (STI) resulting from contamination by the bacterium *Treponema pallidum*. Although syphilis was discovered hundreds of years ago and its form of diagnosis and treatment is widely known, the epidemic grows exponentially in many countries and challenges, mainly, primary health care services [1–3].

According to estimates of the WHO, more than 6 million people aged between 15 and 49 years are infected with syphilis each year worldwide. In addition, the disease is responsible for more than 300,000 fetal and neonatal deaths [1]. In the regions of the Americas, these numbers remain high and impact on health services, especially in countries with limited resources [4,5].

The increase in syphilis in these countries occurs in several population groups, such as pregnant women and newborns. However, the highest rates of the disease are observed in the general population that is not part of these groups, characterizing acquired syphilis [1]. In Brazil, the highest rates of syphilis are also observed in relation to acquired syphilis, making it a neglected group, such as men who have sex with men, sex workers and transgender people.

According to the 2019 Brazilian epidemiological report, acquired syphilis increased from 2.0 cases per 100,000 inhabitants in 2010 to 58.1 cases per 100,000 inhabitants in 2017 [6]. This represents a 2800% increase in the detection rate acquired of syphilis in the country. This growth of the epidemic is influenced by the beginning of late compulsory notification in the general population; however, the structure and organization of primary health care services directly influence disease control strategies [7].

Since syphilis bacterium infection is among the most common STIs, whose transmission occurs mainly through the unprotected sexual act, the actions of condom distribution, the offer of rapid tests for syphilis, adequate treatment and availability of penicillin, are some of the fundamental aspects for combating the epidemic of this STI. In addition to these actions, access, coverage and quality of primary health care may be related to the growth of the disease [8]. Nevertheless, few studies in the literature sought to evaluate the relationship between primary health care indicators, sociodemographic data of the cities and their association with the trends of acquired syphilis.

The significant increase in acquired syphilis shows important flaws in health services and its growth has aroused worldwide concern in recent years. Thus, knowing the trends of

acquired syphilis in Brazil and identifying the factors related to primary health care and the sociodemographic structure of the cities can guide new strategies for health promotion and disease prevention, as well as direct resources that significantly influence the reduction of the epidemic. Thus, the aim of this study is to analyze the trends of acquired syphilis associated with sociodemographic aspects and primary health care in Brazil, in the period from 2011 to 2018.

## Methods

A cross-sectional study was developed to understand the factors related to the trends of acquired syphilis in the period from 2011 to 2019. This study is part of the Brazilian effort to understand the syphilis epidemic and set goals to reduce it, through the project "No Syphilis" whose objective is to reduce acquired syphilis and eliminate congenital syphilis in Brazil [9]. The chosen data collection period is due to the effective notification of acquired syphilis in Brazil, which became compulsory in 2010.

The annual rates of acquired syphilis in the 5570 Brazilian cities were calculated, obtaining them from the number of cases of acquired syphilis (ICD-10: 53.9) in individuals aged 13 years or more in the city, divided by the total local population of individuals in the same age group, multiplied by 100,000. The rates were collected through secondary data from public national health information systems, such as the Notifiable Diseases Information System (SINAN) and the Brazilian Institute of Geography and Statistics (IBGE).

From the annual rates of acquired syphilis, the dependent variable Average Annual Percent Change (AAPC)[10], was calculated, which represents the weighted average of the angular coefficients (W) of the underlying regression line, with the weights equal to the length of the measured interval (b) was calculated. The AAPC is the percentage value that summarizes the variation that occurred during the study period, that is, the number that represents the trend of acquired syphilis from each city.

The independent variables are related to data from Primary Health Care (PHC) from the research of the "National Program for Improvement of Access and Quality of Primary Care (PMAQ)", third cycle. It is a public database, individualized and unidentified, with the objective of evaluating all PHC teams in Brazil. Of the 38,865 teams chosen for the PMAQ survey (S1 Table), teams that did not complete the questionnaire were excluded from the records, totaling 37,350 teams with adequate completion, which is equivalent to a percentage of 96.10%. The responses of the PHC teams were aggregated per cities [11]. The aggregate variables from the PMAQ used in the present study are described in Table 1.

Socioeconomic and demographic variables were collected, such as Human Development Index (HDI), which is the geometric mean of the indexes of the dimensions Income, Education and Longevity; Gini index, which measures the degree of inequality according to per capita household income. Its value ranges from 0, when there is no inequality to 1, when inequality is maximum; proportion of people aged 15 to 24 who do not study, do not work and are vulnerable; proportion of mothers who are wage earners without complete primary education and with children under 15 years of age; total population in permanent private households, categorized into population size. Data related to socioeconomic and demographic municipal characterization were obtained from IBGE.

### Data sources

The AAPC variable that represents the trends of acquired syphilis in each city was categorized: upward, static or downward trend. This theoretical categorization was important due to the characteristic of the AAPC variable, as well as the importance of understanding the factors that

Table 1.  PHC-related variables of the PMAQ.

| Question | Description | Categories |
|---|---|---|
| Rapid syphilis test always available? | Assess the availability of rapid testing in PHC | 1—Partially available<br>2—Fully Available |
| Male condom always available? | Evaluates the availability of male condoms in PHC | 1—Partially available<br>2—Fully Available |
| Female condom always available? | Evaluates the availability of female condoms in PHC | 1—Partially available<br>2—Fully Available |
| Sufficient benzyl penicillin (penicillin G) in PHC | Evaluates the availability of penicillin in PHC | 1—Partially available<br>2—Fully Available |
| Are there families discovered by PHC in the area surrounding the team's territory? | Evaluates PHC service coverage by teams, grouped by municipality | 1 –Partial coverage<br>2 –Fully covered |
| Is benzyl penicillin (penicillin G) applied in the health unit? | Evaluates whether PHC teams apply benzyl penicillin (penicillin G) in the health unit | 1—Not all teams apply<br>2 –All teams apply |
| Proportion of PHC teams with Poor/ Regular grades | Proportion of PHC teams in the municipality with poor or regular grades, obtained from the degree of quality of the teams' work measured in 532 quality standards | 1 –Presence of PHC teams with poor / regular grades<br>2 –No PHC team has poor/regular grades |

influence the upward trend of acquired syphilis in Brazilian cities. The sociodemographic quantitative variables were categorized based on the lowest $p$ value. Then, to identify the association of categorized AAPC between the variables related to the PMAQ and the sociodemographic variables, the bivariate analysis was performed using the Chi-square test. Furthermore, the magnitude of the association was verified through the prevalence ratio for each of the independent variables in relation to the dependent variable, at a significance level of 95%.

To identify the independent factors associated with the AAPC of the upward trend of acquired syphilis, multiple logistic regression analysis was performed. For this, the Stepwise Forward Selection Procedure method was used, in which the variables that compose the model were grouped into blocks, ordering them according to their statistical significance.

The modeling was initiated by the most significant variables of the Chi-square method and, then, the other variables were added one by one, accepting a critical p value of <0.20 to compose the model. The permanence of the variable in the multiple analysis occurred through the likelihood ratio test, absence of multicollinearity, as well as its ability to improve the model through the Hosmer and Lemeshow test. Finally, the residues were analyzed to isolate cases that exert an undue influence on the model, causing little adherence. The data were analyzed using the Statistical Package for the Social Sciences (SPSS) 22.0 program and all analyses considered a significance level equal to 5%.

## Results

From 2011 to 2019, there were 724,310 reported cases of acquired syphilis throughout Brazil. After descriptive analysis of PHC-related variables, the highest proportions are from the partially available rapid syphilis test (52.6%); fully available male condom (85.9%); fully available female condom (62.9%); partially available benzathine penicillin (71.4%); Coverage of PHC teams with partial coverage (47.8%); Application of penicillin in PHC, all teams apply (54.4%); and presence of Poor/Regular teams in PHC (74.1%) (Table 2).

After evaluating the AAPC of acquired syphilis, categorized between upward, static or downward trend, the highest proportion arose in the category of the upward trend, with 87.4%.

**Table 2. Distribution of frequency and proportions of variables related to the basic health service.**

| Category | Frequency | Percentage |
|---|---|---|
| Rapid syphilis test in PHC | | |
| Partially available | 2930 | 52.6 |
| Fully Available | 2382 | 42.8 |
| Male condom in PHC | | |
| Partially available | 529 | 9.5 |
| Fully Available | 4783 | 85.9 |
| Female condom in PHC | | |
| Partially available | 1810 | 32.5 |
| Fully Available | 3502 | 62.9 |
| Benzyl penicillin (penicillin G) in PHC | | |
| Partially available | 3977 | 71.4 |
| Fully Available | 1335 | 24.0 |
| Coverage of PHC teams | | |
| Partial coverage | 2665 | 47.8 |
| Fully covered | 2645 | 47.5 |
| Application of penicillin in PHC | | |
| Not all teams apply | 2282 | 41.0 |
| All teams apply | 3028 | 54.4 |
| Poor / regular grades teams in PHC | | |
| Presence of PHC teams with poor / regular grades | 4128 | 74.1 |
| No PHC team has poor / regular grades | 1177 | 21.1 |

Upon relating the AAPC of acquired syphilis with the PHC-related variables, as well as socioeconomic and demographic variables (Table 3), there was a significant association between the proportion of upward AAPC and the fully available rapid test (88.8%; $p<0.001$); partially available male condom (93.4%; $p<0.001$); partially available female condom (90.1%; $p<0.001$); partially available benzathine penicillin (88.6%; $p<0.001$); partial coverage of PHC teams (92.9%; $p<0.001$); partial application of penicillin in PHC (89.7%; $p<0.001$); higher proportion of women aged 10 to 17 years who had children (88.9%; $p = 0.001$); higher HDI (91.1%; $p<0.001$); higher proportion of teams classified as Poor/Regular in PHC (88.5%; $p<0.001$); and population size with more than 100,000 inhabitants (97.9%; $p<0.001$). There was no significant association between the AAPC and the Gini index and Proportion of persons aged 15 to 24 who do not study, do not work and are vulnerable.

In the logistic regression analysis (Table 4), the following variables remained in the model: not all PHC teams apply penicillin (PRaj = 1.04; $p = 0.002$); higher proportion of PHC teams with poor/regular scores (PRaj = 1.05; $p = 0.001$); population size $>100000$ inhabitants (PRaj = 1.16; $p<0.001$); partially available female condom (PRaj = 1.03; $p = 0.026$).

## Discussion

This is the first research that sought to analyze predictors of the trend of acquired syphilis through PHC and sociodemographic indicators. There were significant associations of poor/regular quality of PHC teams, level of partial coverage of the basic health unit, fully available rapid syphilis test in PHC, partial availability of male condom and female condom, partial availability of benzathine penicillin and not all teams apply penicillin in PHC. Furthermore, there was a significant association of population size with more than 100,000 inhabitants, cities with higher HDI, cities with a higher proportion of women aged 10 to 17 years who had

**Table 3. Comparison of the proportions of the AAPC trends with the variables related to the basic health service and socioeconomic factors.**

| Category | AAPC | | PR | 95% CI | p Value* |
|---|---|---|---|---|---|
| | Growing trend | Static or decreasing trend | | | |
| Rapid syphilis test in PHC | | | | | |
| Fully Available | 2603 (88.8%) | 327 (11.2%) | 1.04 | 1.02–1.06 | <0.001 |
| Partially available | 2032 (85.3%) | 350 (14.7%) | | | |
| Male condom in PHC | | | | | |
| Partially available | 494 (93.4%) | 35 (6.6%) | 1.08 | 1.05–1.10 | <0.001 |
| Fully Available | 4141 (86.6%) | 642 (13.4%) | | | |
| Female condom in PHC | | | | | |
| Partially available | 1631 (90.1%) | 179 (9.9%) | 1.05 | 1.03–1.07 | <0.001 |
| Fully Available | 3004 (85.8%) | 498 (14.2%) | | | |
| Benzyl penicillin (penicillin G) in PHC | | | | | |
| Partially available | 3525 (88.6%) | 452 (11.4%) | 1.07 | 1.04–1.09 | <0.001 |
| Fully Available | 1110 (83.1%) | 225 (16.9%) | | | |
| Coverage of PHC teams | | | | | |
| Partial coverage | 2475 (92.9%) | 190 (7.1%) | 1.14 | 1.11–1.16 | <0.001 |
| Fully covered | 2159 (81.6%) | 486 (18.4%) | | | |
| Application of penicillin in PHC | | | | | |
| Not all teams apply | 2048 (89.7%) | 234 (10.3%) | 1.05 | 1.03–1.07 | <0.001 |
| All teams apply | 2586 (85.4%) | 442 (14.6%) | | | |
| Gini index | | | | | |
| >0.49 | 2391 (87.9%) | 328 (12.1%) | 1.01 | 1.99–1.03 | 0.226 |
| Up to 0.49 | 2472 (86.9%) | 374 (13.1%) | | | |
| Proportion of women aged 10 to 17 who had children | | | | | |
| >2.81 | 2463 (88.9%) | 308 (11.1%) | 1.03 | 1.01–1.06 | 0.001 |
| Up to 2.81 | 2400 (85.9%) | 394 (14.1%) | | | |
| HDI | | | | | |
| >0.665 | 2529 (91.1%) | 246 (8.9%) | 1.09 | 1.07–1.11 | <0.001 |
| Up to 0.665 | 2334 (83.7%) | 456 (16.3%) | | | |
| Proportion of persons aged 15 to 24 who do not study, do not work and are vulnerable | | | | | |
| >32.89 | 2458 (88.4%) | 323 (11.6%) | 1.02 | 1.00–1.04 | 0.025 |
| Up to 32.89 | 2405 (86.4%) | 379 (13.6%) | | | |
| Poor/regular grades teams in PHC | | | | | |
| Presence of PHC teams with poor/regular grades | 3655 (88.5%) | 473 (11.5%) | 1.07 | 1.04–1.10 | <0.001 |
| No PHC team has poor/regular grades | 974 (82.8%) | 203 (17.2%) | | | |
| Population size | | | | | |
| Up to 20000 inhabitants | 3264 (83.3%) | 655 (16.7%) | 1.00 | | <0.001 |
| 20001–50000 inhabitants | 1014 (97.2%) | 29 (2.8%) | 1.17 | 1.15–1.19 | |
| 50001–100000 inhabitants | 313 (96.3%) | 12 (3.7%) | 1.16 | 1.13–1.19 | |
| >100000 inhabitants | 277 (97.9%) | 6 (2.1%) | 1.17 | 1.15–1.20 | |

children and cities with the highest proportion of 15 to 24 years who do not study, do not work and are vulnerable in the population of this age group.

Syphilis is part of PHC-sensitive diseases. Thus, any modification in these services may directly influence its indicators [12,13]. In this study, a higher proportion of PHC teams with poor/regular scores and their significant association with upward syphilis AAPC was demonstrated, evidencing it as an important independent predictor, i.e., every PHC unit with poor/regular teams increases the upward trend of acquired syphilis by 18%.

**Table 4. Independent AAPC-related factors of the growing trend.**

| Reference | Predictor | PR | PRaj | 95% CI | $p^{**}$ |
|---|---|---|---|---|---|
| Application of penicillin in PHC | | | | | |
| All PHC teams apply | Not all teams apply | 1.05 | 1.04 | 1.02–1.6 | 0.002 |
| Poor/regular grades teams in PHC | | | | | |
| No PHC team has poor/regular grades | Presence of PHC teams with poor / regular grades | 1.07 | 1.05 | 1.02–1.08 | 0.001 |
| Population size* | | | | | |
| 20000 inhabitants | >100000 inhabitants | 1.17 | 1.16 | 1.08–2.24 | <0.001 |
| Female condom in PHC | | | | | |
| Fully Available | Partially available | 1.05 | 1.03 | 1.00–1.05 | 0.038 |

* Dummy Variable

** Logistic Regression; Hosmer and Lemeshow = 0.702.

A study developed in Australia by Nattabi et al. [14] demonstrated that the organizational factors of PHC, such as the clinic infrastructure, team profile and its quality can influence the tests and indicators of acquired syphilis. In their multivariate analysis, every improvement in the organizational indicator of the basic health service increased the syphilis indicator in remote populations by 5.1%. In this respect, indicators that measure quality in PHC are important predictors of acquired syphilis and teams classified as unsatisfactory may be contributing to the growth of the epidemic. Thus, the low quality of PHC may be favoring the growth of the acquired syphilis epidemic in Brazil, requiring improvements in its structure and health professionals to reverse the results presented in this study.

In Brazil, the number of families covered by PHC is calculated from territories, and their percentage of coverage between 2006 and 2016 was 45.3% and 64.0%, respectively [15]. Studies that evaluated the relationship between PHC coverage and syphilis show a significant association, such as the study by Nunes et al. [16], which evaluated the correlation between PHC coverage with congenital syphilis and in pregnant women, concluding that the cities with the lowest coverage were related to the higher rates of syphilis detection. Although this study did not address syphilis in the general population and did not test any predictor, cities with a larger number of families discovered by PHC probably influence the trends of acquired syphilis.

In relation to rapid syphilis tests in PHC, studies show them as a fundamental diagnostic triage tool to identify and contain the disease in places with little structure, especially in the identification of people without signs and symptoms, besides being fundamental for the algorithms of definition of cases for treatment [17,18].

In Brazil, the rapid test for syphilis is provided free of charge to the entire population [19], but its lack of supply is a reality and may be contributing to reduced number of diagnoses in PHC and, consequently, to the decreased number of people who should be under treatment. A study conducted in Brazil to verify the implementation of rapid tests in PHC showed that 37% did not have a rapid test in the health unit and 66.6% of the tests had expired [20].

Given the limitation of existing failures in syphilis triage in PHC, the available rapid tests can be rationed and directed only for use in pregnant women during prenatal care, which hinders its use in the general population. Testing the sexual partner and breaking the chain of transmission of the disease are an important challenge, as demonstrated in the study by Cavalcante, Pereira and Castro [21], whose number of partners untreated for syphilis reached 29.8%. Thus, the increase and improvement of management in the universal distribution of rapid tests in PHC can facilitate the triage of syphilis cases in the general population.

Regarding the treatment of syphilis, penicillin remains a highly effective medicine of choice for the individuals infected by *Treponema* [22,23]. The worldwide shortage of penicillin was the subject of the 6th WHO Assembly in Geneva [24], recognizing the shortage of the drug for many years in several countries of the world. In Brazil, it was not different [25]. The shortage of penicillin aggravated the syphilis crisis in the country and caused the Brazilian government to centralize the purchase and make direct distribution to the basic health care units, besides limiting its use only for cases of syphilis [26].

Even after centralized purchase and control in the use and distribution of penicillin, the shortage is still present in the PHC, suggesting that decentralization in drug management would be an important factor to control the epidemic [27,28]. In this sense, the impact of the lack of penicillin in PHC has also influenced the high trends of acquired syphilis in Brazil and the decentralization of its dispensing can improve the effectiveness in the rational use of the drug and reduce its lack in primary care services.

In addition to the availability of penicillin in PHC, there is a high proportion of cities that do not apply penicillin in PHC, favoring the growth of acquired syphilis. A study developed by Araújo et al. [27], which aimed to analyze structural aspects for the prevention of congenital syphilis, found that only 16.9% of the PHC units administered penicillin G, with probable fear of adverse effects of the drug and the lack of structure for its application. Ordinance N. 156, of January 19, 2006 of the Ministry of Health [29], established several criteria to act on adverse reactions in the use of penicillin in PHC, in addition to listing a minimum hospital structure for its use. This ordinance was revoked in 2011, removing these barriers, but it seems that the negative effects and fear of penicillin reactions by health professionals in PHC continues to negatively influence efforts to reduce the syphilis epidemic.

It is noteworthy that a systematic review study attested to the safety in the use of penicillin and did not find any evidence that justifies a hospital structure in PHC. In addition to stressing the safe use of the drug in pregnant women and in the general population, the study encourages that there should not be any political barrier to the use of penicillin in PHC [30]. Thus, the training of health teams to understand the safety in the use of penicillin in PHC can improve their application rates and, consequently, reduce an important barrier for the treatment of acquired syphilis at the site [7].

Another important action developed in PHC is the availability of male and female condoms. A study developed in India using multivariate models to identify coverage and the use of condoms for prevention of HIV and other STIs, in the audience of Men who have Sex with Men (MSM), sex workers and transgender people, demonstrated that the prevalence of reactive serology to syphilis significantly reduced in interventions with condoms, reducing from 8.8% to 1.1% the prevalence of syphilis [31].

In Brazil, a study that evaluated the risk practices of sex workers showed that 40.6% of the people studied did not receive condoms or received insufficient amounts. Although this study did not stratify the effect of social groups on condom use in relation to the upward trend of syphilis, it is likely that these effects will be repeated in the general population, especially if assessing the impact of female condoms on the prevention of syphilis.

The female condom is effective in the prevention of STIs and promotes female autonomy in sexual relations, especially when the partner denies the use of condoms [32]. The use of female condoms still has a low frequency when compared to male condoms, often due to low supply in health services and fear and unawareness of its use [33]. Thus, the promotion of the use of female condoms in PHC can reduce the higher upward trends of acquired syphilis in Brazil.

In relation to sociodemographic variables, studies show that small cities have greater coverage of PHC and, consequently, greater control of cases [34,35]. Naturally, smaller cities have a

higher volume of case detection, due to their oscillation, but when evaluating the population size in upward trends, the larger cities are more important in determining the epidemic, which indicates a greater disorganization of the system against STIs. In relation to HDI, cities with better HDI have a greater difficulty in offering a significant number of prenatal consultations, increasing the probability of acquiring a STI [36].

As demonstrated in the course of this work, syphilis is a PHC-sensitive disease and its structure is fundamental for disease control. Most of the cities with better HDI are in the larger cities, but these do not offer a better equity in health care and syphilis control [37]. It is important to understand that, in relation to syphilis, PHC services are more attended by people with greater social vulnerability, suggesting that cities with better HDI have a greater disorganization in relation to syphilis control policies and that they should be better investigated regarding access to this policy.

Some limitations of this study are related to its nature with secondary data, such as the quality of notifications of acquired syphilis that became compulsory only in 2010, as well as the reliability of national censuses that assess a large volume of variables countrywide. Nonetheless, this study worked with consolidated data of acquired syphilis from 2011. Moreover, it is noticeable the improvement of the quality of the PMAQ records, whose collection is in its tenth year, which can avoid incomplete or inconsistent records by PCH teams. In addition, another limitation to consider is that Brazil is a huge country, with regional socio-demographic inequalities, which are related to the quality of primary care services. This could influence the relationships found in this study, even though these factors have been controlled.

In view of the above, important predictors of the upward trends of acquired syphilis were found in this study, such as the quality of the PHC service, penicillin applied in PHC, if the female condom is fully available and the influence of population size. Thus, measures aimed at improving access and quality of PHC, not only increasing the availability of penicillin, condoms and rapid tests at care places, but also bringing the training of health professionals in the care of STIs, become an indispensable tool, especially in large cities with a higher proportion of young people and young adults in vulnerability. New local studies need to be developed, mainly to understand whether the interventions in PHC suggested in this study can modify the high trends of acquired syphilis in the country.

## Supporting information

**S1 STROBE Checklist. Checklist of items that should be included in reports of *cross-sectional studies.***
(PDF)

**S1 Table. Comparison of the proportions of the AAPC trends with the variables related to the basic health service and socioeconomic factors.** (xlsx).
(XLSX)

## Author Contributions

**Conceptualization:** Marquiony Marques dos Santos, Tatyana Maria Silva de Souza Rosendo, Ana Karla Bezerra Lopes, Angelo Giuseppe Roncalli, Kenio Costa de Lima.

**Formal analysis:** Marquiony Marques dos Santos.

**Funding acquisition:** Angelo Giuseppe Roncalli, Kenio Costa de Lima.

**Investigation:** Marquiony Marques dos Santos, Tatyana Maria Silva de Souza Rosendo, Ana Karla Bezerra Lopes, Angelo Giuseppe Roncalli, Kenio Costa de Lima.

**Methodology:** Marquiony Marques dos Santos, Angelo Giuseppe Roncalli, Kenio Costa de Lima.

**Resources:** Angelo Giuseppe Roncalli, Kenio Costa de Lima.

**Supervision:** Marquiony Marques dos Santos, Angelo Giuseppe Roncalli, Kenio Costa de Lima.

**Writing – original draft:** Marquiony Marques dos Santos.

**Writing – review & editing:** Tatyana Maria Silva de Souza Rosendo, Ana Karla Bezerra Lopes, Angelo Giuseppe Roncalli, Kenio Costa de Lima.

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
