## [Decision Letter · Decision Letter 0]

20 Oct 2020

Dear Dr Marquiony Marques dos Santos

Thank you very much for submitting your manuscript "Weaknesses in primary health care favor the growth of acquired syphilis" for consideration at PLOS Neglected Tropical Diseases. As with all papers reviewed by the journal, your manuscript was reviewed by members of the editorial board and by several independent reviewers. In light of the reviews (below this email), we would like to invite the resubmission of a significantly-revised version that takes into account the reviewers' comments. 

We cannot make any decision about publication until we have seen the revised manuscript and your response to the reviewers' comments. Your revised manuscript is also likely to be sent to reviewers for further evaluation.

Editors' comments: 

- Please present descriptive data in the abstract (number of cases, most important findings from descriptive analysis)

- The presentation of the formula for calculation of AAPC is not necessary - a citation of a reference will be sufficient.

Sincerely,

Jorg Heukelbach

Guest Editor

Olivier Chosidow

Deputy Editor

Reviewer's Responses to Questions

**Key Review Criteria Required for Acceptance?**

**Methods**

-Are the objectives of the study clearly articulated with a clear testable hypothesis stated?

-Is the study design appropriate to address the stated objectives?

-Is the population clearly described and appropriate for the hypothesis being tested?

-Is the sample size sufficient to ensure adequate power to address the hypothesis being tested?

-Were correct statistical analysis used to support conclusions?

-Are there concerns about ethical or regulatory requirements being met?

Reviewer #1: The methods used for the study were acceptable.

Reviewer #2: The objectives of the study are clearly articulated with a clear and testable hypothesis. The study design, as well as the sample (criteria and size) and its time frame are sufficient to meet the stated objectives and for the tested hypothesis.

The analyzes used support the considerations in the present work. The study used secondary, non-collectible ethical data involved.

Although not mandatory, a suggestion that can add even more value to the present study is based on the fact that these data are part of a national project for Rapid Response to Syphilis in Primary Health Care Networks, this project started in Brazil, approximately in 2018, and one of the objectives is to strengthen the fight against syphilis in priority municipalities in Brazil in primary health care, therefore, I suggest that the analyzes include data from 2019, since these actions of supporters in PHC teams can , or not, impact the findings presented until 2018.

**Results**

-Does the analysis presented match the analysis plan?

-Are the results clearly and completely presented?

-Are the figures (Tables, Images) of sufficient quality for clarity?

Reviewer #1: The results are clearly presented; however Table 1, 2 and 3 need to be modified with shading or lines to separate the data presented for each variable. The current presentation makes it difficult for a readers eye to follow the row across.

Reviewer #2: The results are clear and consistent, despite the large number of variables and municipalities involved, the tables and charts are clear and are necessary for understanding the study.

**Conclusions**

-Are the conclusions supported by the data presented?

-Are the limitations of analysis clearly described?

-Do the authors discuss how these data can be helpful to advance our understanding of the topic under study?

-Is public health relevance addressed?

Reviewer #1: The authors concludes are well described ans supported by the data analyzed. 

The authors did a nice job describing how the data illuminate the public health crisis in Brazil and areas which could be improved to help reduce the number of acquired syphilis cases.

Reviewer #2: The conclusions presented are extremely important for coping with Syphilis Acquired in Brazil. 

One of the limitations that the authors do not present is to consider the continental dimension of Brazil, which comprises many sociodemographic differences and inequality in health care coverage in the country, the latter being an important factor in discussing these results.

This study type of study is very important for Public Health, as it relates the service at PHC and the trend of acquired syphilis cases, since in Brazil, it is these teams in primary health care that are at the forefront of coping to that epidemic.

**Editorial and Data Presentation Modifications?**

Reviewer #1: A few minor revisions are required:

1. The author needs to define HDI when it first appear in the document. 

2. The Tables need to be reformatted to allow for easier interpretation by the readers. Addition of shading or lines should improve this issue.

3. If the data are available, it's recommenced that the authors include a table with specifics about the cohort reviewed (e.g. sex, mean age, sexual preference, stage of diagnosed syphilis, etc)

3. On lines 75-77, the authors need to clarify what "group" is they are considering neglected.

Reviewer #2: The suggestion would be to review the data including the 2019 data in the analyzes, as already indicated in the comments on the method.

**Summary and General Comments**

Reviewer #1: The manuscript by Marques dos Santos, et al. supports the idea that modification to Brazil's primary health care systems could help reduces the number of acquire syphilis cases. In general the manuscript is well written and with the minor revisions, this manuscript will help other regions globally to investigate similar factors such as primary health care and sociodemographic structures, and with attention could significantly influence a reduction in syphilis cases.

Reviewer #2: The article discusses an acquired syphilis that has its rates increasing worldwide. This article presents the perspective of the tendency of Syphilis Acquired with Primary Health Care, this look will contribute a lot to the understanding of the increase of an easily treatable and dysagnotic disease, but despite that, its rates continue to increase.

PLOS authors have the option to publish the peer review history of their article (what does this mean?). If published, this will include your full peer review and any attached files.

Reviewer #1: No

Reviewer #2: No
---

## [Editor Report · Decision Letter 1]

22 Dec 2020

Dear Colleague

To:Marquiony Marques dos Santos

We are pleased to inform you that your manuscript 'Weaknesses in primary health care favor the growth of acquired syphilis' has been provisionally accepted for publication in PLOS Neglected Tropical Diseases.

Best regards,

Olivier Chosidow

Deputy Editor

Olivier Chosidow

Deputy Editor

---

## [Editor Report · Acceptance letter]

29 Jan 2021

Dear Dr. Santos,

We are delighted to inform you that your manuscript, "Weaknesses in primary health care favor the growth of acquired syphilis," has been formally accepted for publication in PLOS Neglected Tropical Diseases.

Best regards,

Shaden Kamhawi

co-Editor-in-Chief

Paul Brindley

co-Editor-in-Chief
